# Automated Nanodroplet Dispensing for Large-Scale Spheroid Generation via Hanging Drop and Parallelized Lossless Spheroid Harvesting

**DOI:** 10.3390/mi15020231

**Published:** 2024-01-31

**Authors:** Viktoria Zieger, Ellen Woehr, Stefan Zimmermann, Daniel Frejek, Peter Koltay, Roland Zengerle, Sabrina Kartmann

**Affiliations:** 1Laboratory for MEMS Applications, IMTEK-Department of Microsystems Engineering, University of Freiburg, Georges-Koehler-Allee 103, D-79110 Freiburg, Germany; stefan.zimmermann@imtek.uni-freiburg.de (S.Z.); peter.koltay@imtek.uni-freiburg.de (P.K.); roland.zengerle@imtek.uni-freiburg.de (R.Z.); sabrina.kartmann@hahn-schickard.de (S.K.); 2Hahn-Schickard, Georges-Koehler-Allee 103, D-79110 Freiburg, Germany; ellen.woehr@hahn-schickard.de (E.W.); daniel.frejek@hahn-schickard.de (D.F.); 3Study Program Molecular and Technical Medicine, Faculty Medical and Life Science, University of Furtwangen, D-78054 Villingen-Schwenningen, Germany

**Keywords:** spheroids, high-throughput, 3D in vitro model, spheroid harvesting, large scale

## Abstract

Creating model systems that replicate in vivo tissues is crucial for understanding complex biological pathways like drug response and disease progression. Three-dimensional (3D) in vitro models, especially multicellular spheroids (MCSs), offer valuable insights into physiological processes. However, generating MCSs at scale with consistent properties and efficiently recovering them pose challenges. We introduce a workflow that automates large-scale spheroid production and enables parallel harvesting into individual wells of a microtiter plate. Our method, based on the hanging-drop technique, utilizes a non-contact dispenser for dispensing nanoliter droplets of a uniformly mixed-cell suspension. The setup allows for extended processing times of up to 45 min without compromising spheroid quality. As a proof of concept, we achieved a 99.3% spheroid generation efficiency and maintained highly consistent spheroid sizes, with a coefficient of variance below 8% for MCF7 spheroids. Our centrifugation-based drop transfer for spheroid harvesting achieved a sample recovery of 100%. We successfully transferred HT29 spheroids from hanging drops to individual wells preloaded with collagen matrices, where they continued to proliferate. This high-throughput workflow opens new possibilities for prolonged spheroid cultivation, advanced downstream assays, and increased hands-off time in complex 3D cell culture protocols.

## 1. Introduction

Understanding complex disease pathways, cancer progression, and tissue development is essential for the advancement of modern medicine and drug development. In order to achieve this, physiologically relevant in vitro model systems are needed. Since living organisms are composed of a variety of different cell types growing in 3D within complex architectures of extracellular matrix (ECM) and vasculature, flat 2D in vitro cell culture has a very limited ability to recapitulate in vivo processes [1]. Therefore, 3D cell culture models are becoming increasingly important [2]. In particular, to mimic solid tumors, multicellular spheroids (MCSs) can be used to recapitulate cell–cell and cell–ECM interactions in the primary tissue microenvironment [3,4]. Spheroids are formed by cellular self-assembly in a non-adhesive environment [2]. Because MCSs can mimic key physiological features to model tumorigenesis and tumor progression, it is possible to study complex biological pathways, such as drug response, invasion, and metastasis [3,5].

Various methods to generate MCSs have been reported, such as the hanging-drop method, rotating flasks, ultra-low-attachment plates, and hydrogels that mimic the ECM [6,7]. However, achieving large-scale capability, size and shape consistency, adjustable numbers per well, high harvesting efficiency, time efficiency, and cost-effectiveness for meaningful statistical studies and reproducibility remains a challenge [3]. The hanging-drop method as a scaffold-free approach and the hydrogel-based cultivation of spheroids complement each other in terms of simplicity, process control, long-term culture, and in vivo mimicking microenvironment, which is why we focus on these two methods in this study.

For the hanging-drop method, drops of a cell suspension are dispensed on a substrate that is inverted so that the drops hang downward. Due to gravitational forces, cells accumulate at the lowest point of the cell suspension drops and form dense 3D cell aggregates due to cell–cell contacts. The hanging-drop method is easy to operate and offers automation potential, excellent control over spheroid size, scalability, and shape uniformity [3]. The drawbacks include limited nutrients in the drops and evaporation [8,9], making long-term studies and drug efficacy testing difficult without the use of complex perfusion systems [10,11,12]. In addition, scaffold-free spheroid cultivation in media often lacks an ECM-mimicking microenvironment, affecting the physiological relevance of the model system [13]. While additives like methylcellulose, collagen, or basement membrane matrix can enhance the in vivo mimicking of the microenvironment, cell–cell contact, and spheroid formation [14,15], their inclusion complicates cell suspension handling and dispensing, as they favor nozzle clogging.

Furthermore, Gao et al. showed that flat drop geometries negatively affect spheroid formation, resulting in loosely aggregated cell clusters, whereas only drops with high meniscus curvature result in dense spheroid formation [16]. Commonly observed drop flattening on substrates, caused by surface-modifying proteins in cell culture media, underscores the importance of optimizing the drop geometry and stability [17,18]. Strategies for achieving this optimization include functionalized surfaces with hydrophilic and hydrophobic structures or microfluidic plate designs that limit the wetted area [12,16,19,20,21,22,23]. In addition, surface modifications may also be required for spheroid harvesting, e.g., for endpoint analysis, which would otherwise result in high sample loss [14,21]. While surface modification and microfluidic platform designs yield dense and well-formed spheroids, they require complex preparation steps with non-standard labware and elaborate setups that limit accessibility. Furthermore, automation technologies face challenges such as cell sedimentation, leading to variations in spheroid size and non-robust processes due to nozzle clogging [24].

In contrast to the hanging-drop method, scaffold-based spheroid generation and cultivation approaches provide suitable conditions for long-term culture and drug screening. Scaffold-based approaches rely on the formation of spontaneous cell–cell contacts of cells embedded in hydrogels mimicking the ECM [3,25,26,27,28].

Because cell–ECM contacts can enhance the physiological relevance, scaffold-based cultivation has been shown to improve the proliferation and differentiation of cellular spheroids [29]. In addition, spheroid cultivation in ECM-mimicking hydrogels allows the study of invasion behavior, cell migration, vascularization, and aggregate fusion [30,31,32,33]. The result is a powerful model system for the recapitulation of in vivo tissue properties.

Despite the aforementioned advantages of scaffold-based MSC generation over scaffold-free approaches, automation is complex, and the generated spheroids vary widely in size and shape. Achieving sufficient spheroid size control typically requires many steps and extensive equipment [33,34]. This severely limits process control, standardization, and reproducibility.

Here, we extend our research from [35] and show a combination of scaffold-free MSC generation and scaffold-based MSC cultivation in a novel workflow. In the first step, we use an in-house-developed liquid-handling platform that is optimized for consistent and robust automated spheroid generation via the hanging-drop method. We investigated the large-scale suitability of automated spheroid generation in terms of extended processing times, spheroid size consistency, and spheroid generation efficacy. In the second step, we transfer the mature spheroids in a highly parallel manner without loss from the hanging drops into an ECM-mimicking hydrogel using a standard laboratory centrifuge for extended spheroid culture [36]. We investigated the centrifugal forces required to successfully transfer spheroids into individual wells of a microwell plate and demonstrate highly parallel, loss-free spheroid harvesting with subsequent spheroid proliferation.

Taken together, we combine the advantages of scaffold-free and scaffold-based spheroid cultivation in terms of simplicity, scalability, time efficiency, process control, and in vivo mimicking microenvironment. With that, we enable a high degree of automation and standardization for MCS generation and cultivation.

## 2. Materials and Methods

### 2.1. Cell Culture

MCF7 and HT29 cells were cultured in Dulbecco’s Modified Eagle Medium (DMEM), high glucose, GlutaMAX™ Supplement, and pyruvate (Thermo Fisher Scientific Inc., Waltham, MA, USA) supplemented with 10% fetal bovine serum (FBS, Biochrom AG, Berlin, Germany) and 1% penicillin/streptomycin (Thermo Fisher Scientific Inc., Waltham, MA, USA). Harvested cells were resuspended in 3D Tumorsphere Medium XF (PromoCell GmbH, Heidelberg, Germany) supplemented with 10% FBS and 1% penicillin/streptomycin (complete spheroid medium).

### 2.2. Viability Analysis

Viability analysis was performed by staining cells with Trypan Blue Stain (0.4%) (Thermo Fisher Scientific Inc., Waltham, MA, USA). The number of living and dead cells in a sample was counted with an automated cell counter (Countess II, Thermo Fisher Scientific Inc., Waltham, MA, USA). Viability was calculated as the ratio of counted living cells to total counted cells.

### 2.3. Manual Spheroid Generation

Manual generation of MCSs using the hanging-drop method was performed by dispensing cell suspension drops with an 8-channel pipette (‘Research’ 10 µL, Eppendorf SE, Hamburg, Germany) in a 96-microwell plate (MWP) format into the lid of a 1-well plate (Kisker Biotech GmbH & Co. KG, Steinfurt, Germany). The plate containing the drop arrays was inverted, placed on a reservoir filled with PBS to prevent evaporation, and incubated at 37 °C for four days.

### 2.4. Time–Pressure-Controlled Dispensing

Automated spheroid generation with a commercialized liquid dispenser was also tested for comparison reasons. For that, we applied an I.DOT dispenser (Dispendix GmbH, Stuttgart, Germany) that controls time/pressure to dispense the desired liquid volume. This method has already been used for automated spheroid generation with various cell lines [19,20]. Here, the objective was focused on large-scale spheroid generation as well as the generation of spheroids on standard labware without functionalized substrates. We printed cell suspension drop arrays in a 384-MWP format into the lids of one-well plates and placed them on a reservoir filled with PBS prior to incubation at 37 °C.

### 2.5. In-House-Developed Platform Design and Components for Automated Spheroid Generation

For this study, an in-house-built platform suitable for automated large-scale spheroid generation was developed to produce hanging-drop arrays of cell suspension (see Figure 1a,b). Custom parts of the platform were 3D-printed with a Prusa MK3S (Prusa Research, Praha, Czech Republic) and PETG filament. A standard 15 mL tube containing the cell suspension was placed into the platform. A thin polyimide capillary tube (Zeus Industrial Products, Orangeburg, SC, USA) with an inner diameter of 381 µm and a length of 20 cm was used to fluidically connect the cell suspension reservoir to the droplet dispenser (PipeJet P9, BioFluidix GmbH, Freiburg, Germany). Mixing of the cell suspension in the reservoir was performed using a 5 mL disposable syringe mounted on a motorized linear stage. A PTFE tube with an inner diameter of 4 mm was connected to the syringe at one end and immersed in the cell suspension in the reservoir at the other end. By periodically drawing the cell suspension in and out with the motorized syringe pump at a flow rate of 1.4 to 2 mL/s and a frequency of 0.2 to 0.5 Hz, the cells were kept homogeneously distributed.

A 5MP CMOS camera (IDS UI-3280CP-M-GL R2, Imaging Development Systems GmbH, Obersulm, Germany) with a 6.5X zoom-lens system was integrated into the setup to monitor the free-flying nanoliter droplets ejected by the PipeJet dispenser.

By moving the lid with a motorized xy-stage (Zaber Technologies Inc., Vancouver, Canada), drop arrays of a desired size and arrangement were printed on the lid of a one-well plate. Array patterns matching the layouts of 96- and 384-SBS MWPs were realized, but smaller pitch sizes are also feasible due to the high printing accuracy [37]. To reduce evaporation during the printing process, the substrate on which the drops are printed was cooled to 4 °C with a recirculating chiller (Fryka GmbH, Esslingen, Germany) connected to a heat-dissipating cold plate and a custom-designed aluminum microplate holder. After completing drop-array printing, the lid with the drop array was turned upside down and placed on top of the bottom part of the one-well plate, which was filled with PBS to prevent the evaporation of the drops. The plates with the hanging-drop arrays remained in the incubator at 37 °C for three to four days.

### 2.6. Droplet Volume Calibration

The volume of the ejected fluid was determined via computer-based image processing, as described in detail in [38,39]. In brief, images of free-flying droplets were acquired with a CMOS camera. Background subtraction of these images was performed, and the drop contour was determined. Then, each pixel row of the drop was treated as a cylinder, and the volume of each cylinder stack was summed to estimate the total volume of a droplet. Depending on the used PipeJet parameters, droplet volumes between 40 nl and 100 nl were measured. With the previously determined volume $V_{D}$ of one ejected droplet, it was possible to identify the number n of droplets needed to print the desired volume $V_{HD}$ for one hanging drop into the lid of a one-well plate:(1)$$n=\frac{V_{HD}}{V_{D}}$$

### 2.7. Contact Angle Measurements

Contact angles between substrate and spheroid medium were used to investigate the wetting behavior of the used complete spheroid medium on the substrate. Throughout the work presented, hanging-drop arrays were printed into the lids of untreated polystyrene one-well plates. We used the needle method to measure the contact angle between the polystyrene surface and the complete spheroid medium. In brief, a needle with an inner diameter of 510 µm is used to gradually increase the volume of a drop on the surface. With increasing drop volume, the advancing contact angle and the drop baseline are measured. After dispensing 30 µL, the drop is allowed to rest for a few seconds to determine the static contact angle. Then, the needle is used to withdraw liquid again, and the receding contact angle is measured.

In addition, 2 µL drops were dispensed using different dispensing methods (manual pipetting, I.DOT, PipeJet) to measure the static contact angle between the drop and the substrate in relation to the dispensing dynamics.

For the contact angle measurement, we used a contact angle goniometer (OCA 15 plus, Dataphysics Instruments GmbH, Filderstadt, Germany) and the corresponding software (SCA20, Version 5).

### 2.8. Spheroid Harvesting

For harvesting mature spheroids, the target wells of a standard 384-MWP were preloaded with 30 µL of 3 mg/mL collagen (Collagen I, High Concentration, Rat Tail, Corning Inc., New York, NY, USA). The MWP was kept at 4 °C at all times. The plate with the hanging-drop array was then placed above the preloaded 384-MWP and put into a precooled centrifuge (Heraeus Multifuge 3SR+, Thermo Fisher Scientific Inc., Waltham, MA, USA). The hanging drops containing the formed spheroids were centrifuged into the well plate according to the drop volume and geometry with the necessary centrifugal force. The microwell plate with the transferred spheroids was placed into the incubator to allow hydrogel polymerization. Afterward, the hydrogel pellet was covered with 60 µL of the complete spheroid medium. The growth of the spheroids in the hydrogel was observed afterward with a 4× light microscope.

### 2.9. Data and Image Analysis

Generated spheroids were imaged with 4× or 10× magnification of an inverted light microscope (CKX41, Olympus K.K., Shinjuku, Japan). The area A of the spheroids was determined with custom-written Python scripts using the scikit-image packages [40], and the diameter d was estimated as follows:(2)$$d=2\cdot \sqrt{A/\pi }$$

### 2.10. Data Presentation and Statistical Analysis

Values state the mean values ± standard deviation. Box plots extend from the first to third quartile of the data, and the median is shown as the centerline. The whiskers extend from the edges of each box to the last data point in the 1.5× interquartile range, and outliers are shown as circles. Bar and error bar plots show the mean and the standard deviation with error bars.

For statistical assessment, we used a two-sided *t*-test for data sets, for which a normal distribution was tested and assumed with the D’Agostino–Pearson test for normality. For non-normally distributed data, we used the two-sided Mann–Whitney U-test. *p* values  ≥  0.05 were considered not significant (n.s.), *p* < 0.05 is *, *p* < 0.001 is **, and *p* < 0.0001 is ***.

## 3. Results and Discussion

### 3.1. Automated Generation of Hanging-Drop Arrays for Large-Scale Spheroid Generation

Our liquid-handling platform is based on the nanoliter dispensing of a cell suspension performed with a non-contact piezoelectric dispenser (PipeJet P9, BioFluidix GmbH, Freiburg, Germany). In several former studies, we successfully exploited the used dispensing technology in a number of 3D bioprinting applications using different cell types [33,34,41]. Compared to other dispensing techniques, the nanoliter dispenser has a low impact on living cells and highly accurate droplet ejection. For this study, we specifically optimized it for cell suspension printing and the hanging-drop method, as shown in Figure 1a,b. A reservoir with the cell suspension is placed into the platform, and a thin polyimide capillary with an inner diameter of 381 µm is inserted into the reservoir with one end and clamped into the PipeJet dispenser with the other end. Droplets of the cell suspension are ejected by squeezing the tube due to the movement of the piezo-stack actuator of the PipeJet [42]. Capillary forces lead to the constant refilling of the capillary with the cell suspension. 

The volume of suspension droplets dispensed depends on the preset dispenser parameters and can be measured with an integrated camera. The target substrate is then mounted 4 mm below the dispensing nozzle, and multiple nanoliter droplets are dispensed onto the same target spots to create microliter-sized drops for spheroid formation (Figure 1c). With a motorized xy-stage, the generation of any desired drop array pattern is possible. In order to maintain a homogeneous cell suspension at all times during automated dispensing, the cell suspension in the reservoir tube is gently mixed. The motorized syringe pump continuously aspirates and dispenses approximately 30% of the suspension in the reservoir at 1.4 mL/s to 2 mL/s through a 4 mm inner diameter PTFE tube. In this way, it is possible to prevent the formation of a cell pellet in the reservoir tube even after long processing times. The maximum shear stress *τ* on the cells during mixing can be estimated with
(3)$$\tau =\mu \frac{Q}{\pi R^{3}}$$
where μ≈0.9 mPas is the dynamic viscosity of water, Q is the maximum flow rate, and R is the diameter of the mixing tube. The maximum shear stress created through the mixing or the flow in the aspiration capillary does not exceed 80 mPa. The shear stress applied is well below the critical regime of 750 mPa for adherent mammalian cells, where effects on the cells would be expected [43], and is therefore also not considered critical for cell dispensing. To further confirm that mixing and dispensing did not negatively affect the cells, a viability assay was performed with Trypan Blue after mixing cells of the breast cancer cell line MCF7 for 20 min and dispensing them. The viability of the mixed cells, with (94.7 ± 2.0)%, and the viability of the mixed and dispensed cells, with (95.1 ± 6.5)%, were not significantly different from the control samples, with a viability of (95.0 ± 4.4)% (Figure 1d). 

To initiate spheroid formation, the substrate with the cell suspension drop arrays is inverted so that the drops hang downward. This causes the cells to accumulate at the lowest point of the drop meniscus (Figure 1e) and to form dense aggregates after three to four days in the incubator. 

For robust spheroid formation, it is necessary that the hanging drops have a stable, highly curved meniscus, as this favors spheroid aggregation. Cell culture media contain many different components, such as proteins, that have a wetting effect. This is particularly problematic for untreated or unfunctionalized substrates, as it can cause dispensed drops to spread out and flatten on the surface. We therefore measured the static contact angle and, in addition, the advancing and receding contact angles with the needle method for the used complete spheroid medium (Appendix A). A strong hysteresis of the advancing and receding contact angle was measured. As soon as a certain area on the substrate came into contact with the medium, wetting occurred over this entire area, even if the volume decreased again.

We also compared the contact angles of 2 µL drops dispensed by manual pipetting, PipeJet (PJ) dispensing, and time–pressure (TP) dispensing with the I.DOT (Appendix A). We observed that the dispensing method has a high impact on the resulting drop shape (Appendix A). Pressure-based dispensing leads to a more flattened drop shape. The drop created by PJ dispensing shows contact angles closest to the measured static contact angle and therefore has the highest curvature. We believe this is due to the multiple doses per drop and the very confined area in which the ejected PipeJet nanoliter droplets impact the substrate, limiting the wetted surface area [37].

In general, a short drop baseline with a high contact angle is considered favorable for spheroid generation via the hanging-drop method [16]. In this scenario, the effects of drop flattening due to evaporation are decreased, and spheroid formation is stabilized. We also found that satellites were reduced with PJ-based droplet dispensing because the nozzle was in close proximity to the target surface (see Appendix A). In summary, the generation of drop arrays on untreated substrates using a PipeJet shows advantages over manual pipetting and pressure-based dispensing with respect to a drop geometry suitable for spheroid formation. 

### 3.2. Automated Large-Scale Generation of Consistent and Scalable Spheroids

Many automated liquid-handling platforms suffer from nozzle clogging or cell sedimentation as the processing time progresses. This leads to limited sample-processing times, low sample quality or low sample consistency, and thus, unsuitability for large-scale use. In order to create uniformly shaped and sized spheroids, it is essential that the number of cells per drop stays constant. This requirement is met by a homogeneous cell suspension and the accurate and constant dispensing of the hanging-drop volume. A homogeneous cell suspension is achieved due to mixing in the reservoir. In addition, we have also taken into consideration that the ejected droplet volume depends on the filling level of the reservoir. To this end, we have implemented an automatic correction of the number of nanoliter droplets ejected to form a hanging drop of the desired volume with respect to the filling level (for more details, see Appendix A).

In order to assess the large-scale suitability of spheroid generation with PJ-based dispensing and reservoir mixing, we generated drop arrays over extended periods of time and investigated whether this affected the resulting spheroid formation. 

We tested three different approaches in total to compare our results: We generated hanging-drop arrays with our approach based on PJ dispensing with a mixed-cell suspension; we generated hanging-drop arrays by manual pipetting with a multichannel pipette; and we used an I.DOT with TP dispensing that was recently used for spheroid generation via hanging drop in the literature [19,20]. In all cases, we processed the MCF7 cell suspension with a concentration of $3.2\times 10^{4}$ cells per milliliter and generated the arrays on non-functionalized standard microwell plate (MWP) lids. For PJ- and TP-based printing, a total of 1152 drops with a volume of 2 µL each were dispensed in the format of three 384-MWP arrays. For the manual generation of hanging-drop arrays, we used an eight-channel pipette and created one 96-MWP array with 2 µL drops. We repeated our experiments for three different passages.

In general, the PJ-based dispensing process was robust, and no nozzle clogging occurred within 45 min of processing. User interaction during this time was limited to changing the target plates for the drop arrays. In contrast, TP-based dispensing without mixing typically stopped after about 6 min due to nozzle clogging. This is also reflected in the drop generation success efficiency, which describes the ratio of successfully printed drops to the target number of generated drops (Figure 2a). While the drop generation efficiency for PJ-based dispensing remained above 97% even at long dispensing times, TP-based dispensing resulted in insufficient drop generation after about five minutes.

After four days of incubation, the hanging drops were examined to measure the spheroid generation efficiency as the percentage of successfully formed spheroids relative to the target number of drops. PJ dispensing resulted in a spheroid generation efficiency of (99.3 ± 0.9)% for cell suspension processing times of 45 min. Compared to manual pipetting with a spheroid generation efficiency of (98.3 ± 2.2)%, the consistency of successful spheroid generation was slightly improved for PJ-based generation. It should be noted that, although the target substrate is cooled during the printing process, the sample reservoir of the platform is kept at room temperature and in a normal atmosphere. Therefore, the cells are not maintained under optimal conditions. This means that changes in cell proliferation or spheroid formation are more likely the longer the cells are kept outside the incubator. For this reason, we have not tested processing times longer than 45 min. However, processing times could be easily extended from the technical side, as further customization of the platform with atmosphere and overall temperature control is possible to allow for longer dispensing times with enhanced cell environment conditions.

In strong contrast, TP-based cell suspension printing performed spheroid generation with efficiencies of only (79 ± 28)%, which is also very sensitive to the printing time interval of the total suspension processing time (Figure 2a). This is mainly related to the lower efficiency of drop generation due to nozzle clogging. However, we also observed a slightly reduced spheroid generation efficiency in the peripheral areas of the arrays. Drops at the edge of the well plate are more likely to evaporate due to increased air circulation. Especially for the TP-generated drop arrays with wide drop baselines, this results in very flat drops in the peripheral areas of the well plate lid, which we associate with the lower spheroid generation efficiency in this area.

We next analyzed the resulting diameters of the generated spheroids to determine whether the processing and dispensing time had a measurable effect on the resulting spheroid size properties (Figure 2b). For PJ-based dispensing, spheroid size did not change with the increasing processing time of the cell suspension. Due to the mixing of the cell suspension in the reservoir and the droplet volume correction for the decreasing fill level, the volume of hanging drops, as well as the cell concentration in the dispensed drops, remained constant, and thus, the resulting spheroid diameters had a coefficient of variation (CV) of (7.4 ± 0.2)%. This is also evidenced by the significant reduction in the tails of the size distribution compared to the other investigated spheroid generation methods.

In contrast, TP-based cell suspension printing resulted in a larger CV of the spheroid diameter of (21.7 ± 0.9)%. Thus, a variety of different-sized spheroids were present. In TP-based printing, the reservoir is located above the dispensing nozzle. At the beginning of the dispensing process, cells are homogeneously distributed, and spheroid sizes similar to those obtained with manual pipetting are generated (0–2 min in Figure 2b). However, after a short time without mixing in the reservoir, the cells sediment, and a non-homogeneous cell distribution is manifested in the reservoir, with a higher cell density closer to the nozzle and a lower cell density on the upper layers of suspension in the reservoir. As dispensing continues, the high-cell-concentration layer is dispensed, resulting in either nozzle clogging and low drop generation efficiency (Figure 2a) or an enlarged tail toward larger spheroids (2–4 min in Figure 2b). After the high-cell-concentration layer is dispensed, only the low-cell-concentration suspension remains in the reservoir, resulting in smaller spheroid sizes (4–6 min in Figure 2b).

In general, only PJ-based spheroid generation showed no significant differences compared to manual spheroid generation, even for long processing times. This demonstrates the success of our platform in automating the entire process and making it suitable for large-scale production.

The size of the resulting spheroids depends on the number of initial cells per drop. This can be easily controlled by either adjusting the cell concentration of the suspension or by changing the volume of the hanging drop. Since the latter requires fewer manual steps, we tested the scalability of the spheroid size on our platform using MCF7 cells as an example by keeping the cell concentration constant and varying the hanging-drop volumes. Different volumes (1–4 µL) of hanging drops were dispensed, and the spheroid diameter was analyzed after 4 days of incubation (Figure 2c,d). In this volume range, we did not observe a drop-volume-dependent effect on drop printing and spheroid generation efficiency. Out of 15 targeted spheroids, 100% spheroid generation efficiency was observed for each hanging-drop volume. For hanging drops ≤ 1.5 µL, it is advantageous to add empty hanging drops (e.g., water) at the drop array boundaries to reduce evaporation effects. Further, we observed a clear increase in spheroid size with increasing cell-seeding density per drop.

### 3.3. Spheroid Harvesting via Centrifugation

Since nutrients are limited in hanging drops and medium exchange requires complex setups and perfusion systems [12,16,21,22,23], the hanging-drop method is not suitable for long-term culture or drug screening per se. Therefore, it may be necessary to harvest spheroids, transfer them into hydrogels that mimic the extracellular matrix, and support them with additional cell culture medium. However, spheroid harvesting is one of the major causes of high sample loss [14]. Direct generation of spheroids within hydrogels presents difficulties in spheroid size scalability, results in high CVs of the resulting sizes, and involves many manual, non-standardized steps. To combine the size scalability and automation feasibility of the hanging-drop method with the possibility of long-term cultivation and the physiological relevance of hydrogel culture, we established a simple workflow for spheroid recovery using a standard laboratory centrifuge (Figure 3a) [36]: Spheroids are generated in hanging-drop arrays in standard MWP formats on the inside of an MWP lid. After successful spheroid formation, the lid with the hanging drops is placed onto a target MWP that is preloaded with hydrogel and centrifuged. The applied centrifugal force transfers the spheroids from the hanging drops into the hydrogel of the well below.

The centrifugal force required depends on the volume, density, and geometry of the hanging drop. To detach the hanging drop containing the spheroid from the plate, the centrifugal force must exceed the surface tension of the drop [16]:(4)$$\rho a_{C}V\geq 2\pi r\gamma \sin\theta$$
where $a_{C}=R\omega^{2}$ is the centrifugal acceleration, with ω being the angular velocity of centrifugation and R being the radius of the circular movement; V is the drop volume; ρ is the density of the culture medium; r is the radius of the drop contact line on the target plate; γ is the surface tension of the liquid–air interface; and θ is the contact angle between the drop and the plate (Figure 3b). With Equation (2), the required centrifugal acceleration $a_{C}$ for detachment is described as follows:(5)$$a_{C}\geq \frac{2\pi r\gamma \sin\theta }{m}$$
where m is the mass of the drop. Equation (5) shows a static situation with a fixed drop shape. However, it is important to note that as centrifugation begins, the drop shape changes, and a dynamic situation is created. Nevertheless, Equation (5) can be used to estimate the centrifugal force required for successful drop detachment. The centrifugal force has always been chosen to be as low as possible to minimize cell damage. For drops with a volume ≥ 1 µL, the required centrifugal acceleration is <40 g, which has also been verified experimentally.

The donor plate with the hanging-drop array and the receiver plate travel in a circular orbit in the centrifuge. The centrifugal force acting on the drops and, thus, also their movement vector, once detached, is perpendicular to the axis of rotation and always directed radially outwards (Appendix A). Therefore, the impact position of the drops on the receiver plate is shifted with regard to their source position on the donor plate (Appendix A). We measured the drop deflection angle α by measuring the drop deflection length d on the receiver plate with regard to the height h between the donor and receiver plates (Figure 3c). For this investigation, h was 11.8 mm, and the receiver plate was an unstructured one-well plate on which 77 hanging drops with 2 µL each were equally distributed (Appendix A). With the obtained angle, it was then possible to determine the absolute deflection at the level of the well opening of a 384-MWP plate, where the distance h between the donor plate and the well opening is around 0.8 mm (Figure 3d). The drop deflection for drops located at the rear end of the MWP with respect to the direction of rotation is significantly increased due to the circular motion of the plate and the radially outward trajectory of the drops. However, the deflection at the level of the well opening remains below 1 mm and is therefore small enough for all drops to enter a 384-MWP well, for which the radius of the well opening is 1.55 mm.

The transfer efficiency of spheroids from the hanging drops into the wells of a 384-MWP was tested with MCF7 spheroids cultivated in hanging-drop volumes ranging from 1 µL to 4 µL, as well as with spheroids formed from the colon cancer cell line HT29 and cultivated in 3 µL drops. The spheroid diameters ranged from 80 µm to 230 µm at the time point of centrifugation. Centrifugal acceleration was chosen according to Equation (5) with the respective drop volume and ranged from 25 g to 40 g.

Within one minute, centrifugation transferred (100 ± 0)% of the spheroids from the hanging drops to the well below (N > 3, n > 140). This represents a dramatic increase in the harvesting efficiency with minimized manual labor compared to previously reported efficiencies of only 54.3% [14].

Although centrifugation transferred 100% of the spheroids from the hanging drops to the well below, the initial position of the drop on the well plate is critical for the deflection and thus the final position of the spheroid within the well. Spheroids from drops located at the rear end of the MWP with respect to the direction of rotation tend to stick more often to the walls of the wells of a 384-MWP, where they are difficult to image. Either larger well diameters are required or the last three rear wells should not be seeded in order to obtain high-quality observations under the microscope.

In addition, we investigated HT29 spheroid proliferation after centrifugation into 3 mg/mL collagen. Spheroids were generated in 3 µL drops, with 400 cells per drop, in a 384-MWP format. After three days of incubation, the plate containing the mature spheroids was placed like a lid on the preloaded 384-MWP, and the assembly was centrifuged at 25 g for 1 min. After transfer into the collagen matrix, HT29 spheroids were observed for several days (Figure 3e,f). Spheroids were kept in culture for >7 days, during which proliferation and growth were observed. After one day in the hydrogel, the spheroid size shrank compared to the initial size. We associate this with the densification of the cells within the aggregate due to the cell–matrix contacts that are established. After day 1, the spheroid size started to increase, proving the viability and proliferation of the cells.

In general, we conclude that the centrifugation of spheroids from hanging drops into target wells is a highly parallel harvesting method that maximizes sample recovery and minimizes the manual workload. We did not observe spheroid fragmentation as long as the hydrogel was not polymerized. Since centrifugal forces are kept low and act on the spheroids only for a short time, the impact on spheroids is not considered critical compared to standard protocols where spheroids are directly pipetted and mixed in hydrogels. Taken together, the highly parallel spheroid recovery provides very favorable conditions for standardized endpoint assays, drug screening, co-culture, or the long-term cultivation of MCSs. This significantly enhances the ability to design and perform such studies in high throughput compared to standard ultra-low-attachment plates.

## 4. Conclusions

Our platform represents a powerful tool for well-controlled, large-scale spheroid generation. We demonstrated a high degree of automation, where user interaction was limited to exchanging the target substrates. Our system with integrated cell suspension mixing and low-impact nanoliter-drop printing showed superior robustness and quality for spheroid generation as compared to pressure-driven dispensing from a nozzle or manual pipetting. It was not necessary to supplement the spheroid culture media with additives to promote spheroid formation. In addition, we were able to produce stable drop geometries in any desired array formats on non-functionalized commercial standard lids of MWPs, significantly reducing the cost of consumables. All parts that come into contact with the cell suspension are commercially available disposable products and easily replaceable, eliminating the need for costly and complex microfluidic devices and the risk of cross-contamination. Even for prolonged processing times of hanging-drop arrays, high spheroid formation efficiencies were observed, and the resulting spheroids had high size consistency. Control of the spheroid size could easily be realized by adjusting the hanging-drop volume and thus the total number of cells per drop.

We further presented and characterized a highly parallel workflow for spheroid recovery from hanging drops. A simple centrifugation step enables the simultaneous transfer of spheroids into an MWP that can, for example, be preloaded with an ECM-mimicking environment. We showed that only low centrifugal acceleration is required to successfully transfer 100% of spheroids from hanging drops in the wells of a 384-MWP. Spheroids could be embedded in one step into a collagen hydrogel and proliferated for multiple days. This depicts an important step toward the automation, parallelization, and standardization of 3D cell culture models.

Within this work, we combined the benefits of scaffold-free and scaffold-based 3D in vitro cell culture in terms of automation, physiological relevance, standardization, scalability, and reproducibility. Previous drawbacks of the hanging-drop method, namely, labor-intensive procedures, difficulties in long-term culture and spheroid transfer, and low spheroid yields, have been specifically addressed. Our combined workflow of hanging-drop spheroid generation and highly parallel spheroid transfer by centrifugation offers substantial advantages over other commonly used spheroid generation techniques (Table 1). Although there is a considerable one-time cost for the platform, the ongoing cost of disposables is kept low by the ability to use standard labware, increasing the cost-effectiveness of large-scale spheroid production. We are therefore contributing to more reliable and meaningful 3D in vitro models with simplified generation protocols. Eliminating complex, expensive microfluidic designs, perfusion systems, and functionalized substrates in favor of a simple nanoliter dispenser, standard labware such as MWPs, and a lab centrifuge increases accessibility and adaptability to many different workflows. This poses new opportunities for advanced drug screenings, disease pathway modeling, tissue engineering, and functional analysis.

## 5. Patents

The transfer of particles such as spheroids from hanging drops to a receiver vessel is registered as a patent by Sabrina Kartmann, Kevin Tröndle, Peter Koltay, and Csaba Jeney (EP 3 815 788 A1).

## Figures and Tables

**Figure 1 micromachines-15-00231-f001:**
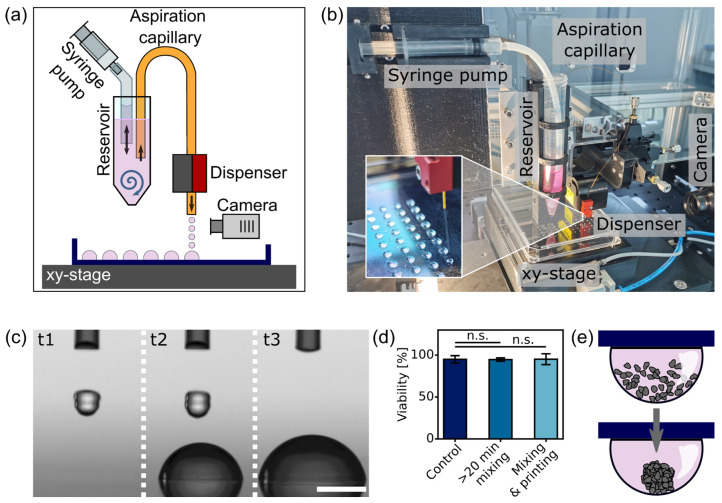
Automated generation of hanging-drop arrays: (**a**) Scheme of the cell suspension printer. Nanoliter droplets are dispensed with a non-contact piezoelectric dispenser (PipeJet P9, BioFluidix GmbH, Freiburg, Germany) onto a substrate. A homogeneous cell distribution in the reservoir is achieved with a motorized syringe pump. In order to generate custom drop arrays, the substrate is moved with a motorized xy-stage. The inlet shows the droplet printer ejecting nanoliter droplets onto a target plate. (**b**) Photograph of the platform. All components in contact with the cell suspension are standard disposable labware. (**c**) To produce hanging drops of accurate volume, the volume of the free-flying nanoliter droplets is first measured. A predefined number of nanoliter droplets are then dispensed onto the same location to create microliter-sized drops on the substrate. (**d**) Analysis of cell viability after processing of the cell suspension compared to a control sample. *p* values  ≥  0.05 were considered not being significant (n.s.). (**e**) The principle of spheroid formation in a hanging drop. Cells accumulate at the lowest point of the drop and form dense cell aggregates after a couple of days.

**Figure 2 micromachines-15-00231-f002:**
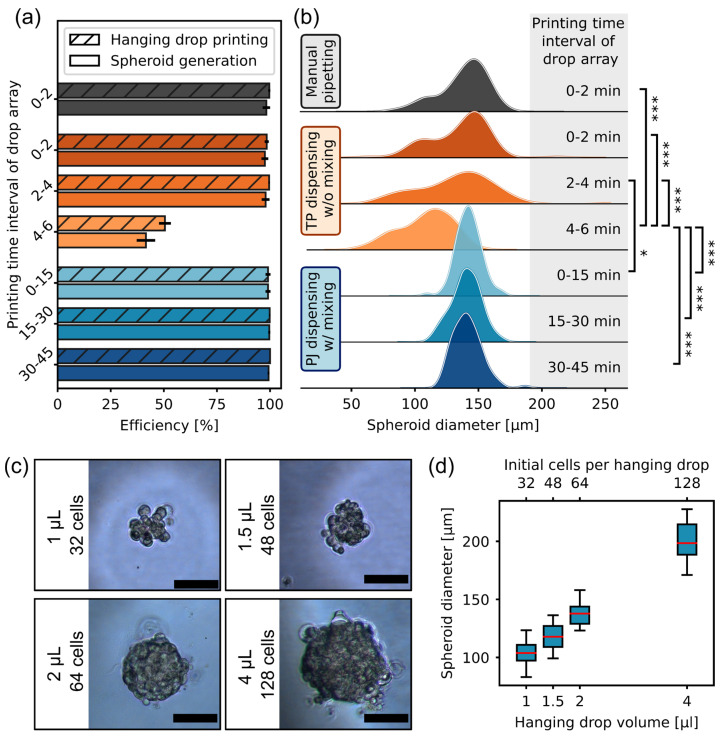
Generation of consistent and scalable spheroids from the breast cancer cell line MCF7: (**a**) Efficiency of the drop printing process and spheroid generation with regard to the printing time interval of the hanging-drop array (TP- and PJ-based dispensing: N = 3, n = 3456; manual pipetting: N = 3, n = 288). (**b**) Evaluation of MCF7 spheroid sizes generated after 4 days of incubation. For TP- and PJ-based dispensing, the diameter of 90 randomly selected spheroids was determined from 1152 drops dispensed within different time intervals (N = 3, n = 270). For comparison, the diameters of 30 randomly selected spheroids were determined from 96 manually pipetted drops (N = 3, n = 90). *p* values  ≥  0.05 were considered not being significant and were not marked, *p* < 0.05 is * and *p* < 0.0001 is ***. (**c**) Exemplary images of formed spheroids 4 days after drop printing depending on the hanging-drop volume and the initial number of cells per drop. Scale bar: 100 µm. (**d**) Scalable size of MCF7 spheroids formed after 4 days. With a seeding cell suspension concentration of $3.2\times 10^{4}$ cells per milliliter, different hanging-drop volumes and thus different cell numbers per drop were realized (n = 15).

**Figure 3 micromachines-15-00231-f003:**
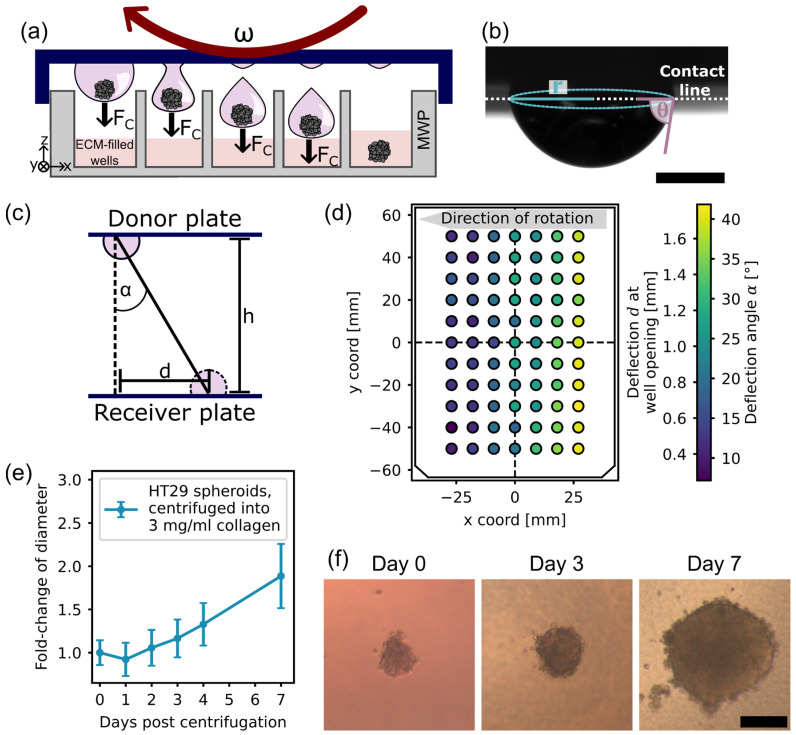
Highly parallelized spheroid harvesting via centrifugation: (**a**) Schematic illustration of centrifugal detachment of a hanging drop containing a spheroid. Spheroids can be simultaneously transferred into individual wells preloaded with an ECM-mimicking hydrogel, for example. (**b**) Example of a hanging drop with 2 µL volume. The drop geometry, defined by the contact-line radius r of the drop with the substrate and the contact angle θ between the liquid and the substrate, determines the surface tension force that must be overcome by the centrifugal force for the drop to detach. Scale bar: 1 mm. (**c**) Schematic illustration of drop deflection due to the Coriolis force that appears as a consequence of the donor and receiver plates rotating on a circular path. (**d**) Measurement of the absolute deflection and the deflection angle of 2 µL drops across an MWP due to the Coriolis force at a centrifugal acceleration of 25 g. The explanation for the position dependence of the deflection angle is provided in supplement Appendix A. (**e**) Growth of spheroids from the colon cancer cell line HT29 that were centrifuged from hanging drops into individual wells of a 384-MWP preloaded with 3 mg/mL collagen (n ≥ 30). (**f**) Example of an HT29 spheroid centrifuged from a hanging drop into a collagen matrix. Day 0 was the day of centrifugal transfer from the drop into the hydrogel. Scale bar: 100 µm.

**Table 1 micromachines-15-00231-t001:** Comparison of the presented spheroid generation method via hanging drop and controlled centrifugal transfer into a hydrogel with other commonly used spheroid generation techniques [2,3,6,44,45,46,47]. We have rated the above methods on a scale from − −, which indicates low suitability, to + +, which indicates high suitability of the method to fulfill a given criterion regarding spheroid generation. ‘o’ indicates moderate suitability of the method to fulfill a given criterion regarding spheroid generation.

	Hanging Drop and Centrifugal Transfer into Hydrogel	Non-Adhesive Well Plates	Hydrogels	Rotating Flasks	Microfluidic Chips
Easy to operate	++	++	+	++	− −
Large-scale generation	++	++	+	++	− −
Spheroid size scalability	++	−	−	− −	++
Spheroid size uniformity	++	++	− −	− −	++
In vivo mimicking microenvironment	+	−	+	− −	++
Long-term culture	++	++	++	++	++
Labor saving	++	− −	−	++	− −
Yield	++	++	+	++	++
Controlled quantity	++	++	−	− −	++
Cost-efficient and simple consumables	o	−	+	++	− −

## Data Availability

The data presented in this study are available on request from the corresponding author.

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
