# Peer review of "Automated Nanodroplet Dispensing for Large-Scale Spheroid Generation via Hanging Drop and Parallelized Lossless Spheroid Harvesting"

_micromachines, 2024, doi:10.3390/mi15020231_

Round 1
Reviewer 1 Report
Comments and Suggestions for Authors
The authors have presented an innovative original research article regarding biomanufacturing. The article is very well-written. The only recommendation i make is to include further information on how this technology features against other biomanufacturing technologies.
Author Response
We sincerely thank the reviewer for the positive feedback and encouraging remarks. We have now added a table in the Conclusion chapter that compares our novel workflow with common alternative technologies for spheroid generation reported in the literature, such as rotating cell culture flasks, ultra-low attachment plates, hydrogels, or microfluidic chips. The disadvantages of the hanging drop method reported in the literature, namely difficulties in large-scale production, difficulties in long-term culture, high workload and therefore low yield, as well as difficulties in transferring the spheroids, are clearly addressed in our combinatorial workflow of cell suspension mixing during automated hanging drop generation and centrifugal spheroid transfer. This provides significant advantages over other techniques reported in the literature to date. Furthermore, user-driven workflow customization can be easily performed in a standardized and automated manner. This allows for a broad versatility of the presented method for a wide range of applications.
Reviewer 2 Report
Comments and Suggestions for Authors
This is a nice and very timely study of droplet generation and spheroid harvesting. This technique should be of interest to a number of researchers in the organic field. I see no issues with this manuscript.
Author Response
The reviewer's valuable feedback is sincerely appreciated. We were very pleased about the enthusiastic and positive assessment of our manuscript.
Reviewer 3 Report
Comments and Suggestions for Authors
The article presents a liquid handling platform for automated large-scale spheroid generation and a simple centrifugation step enabling the simultaneous transfer of spheroids into an MWP. The logic of the paper is clear and the experimental content is substantial. The article can be considered for publication in " Micromachine " after revising the following questions. The comments are below.
1) The introduction expounds on the MCS generation method and spheroid harvesting in detail, but the focus needs to be more prominent. The brief introduction to the work is also lengthy, which may confuse the reader.
2) How long can the PJ-based dispensing process keep working with no nozzle clogging occurring? The authors only conducted the test for 45 minutes, but the longer test is also valuable.
3) The authors should provide the specific efficiency of spheroid generation with different volumes.
4) The authors expound on spheroid harvesting via centrifugation with an example of a 2 µL hanging drop. How effective is this method for other volumes of spheroids?
5) In order to visualize the performance of the platform, the author should provide a table comparing the differences between this method and other commonly used methods in terms of production efficiency, dimensional accuracy, cost, etc.
Author Response
We sincerely appreciate the reviewer's comments and constructive feedback, which clearly helped to improve the quality of the paper. We have addressed the comments point by point below and highlighted the changes in the manuscript in red. We hope that all concerns have been sufficiently addressed in our revisions.
- The introduction expounds on the MCS generation method and spheroid harvesting in detail, but the focus needs to be more prominent. The brief introduction to the work is also lengthy, which may confuse the reader.
Response: We express our appreciation to the reviewer for the constructive and valuable remark. We restructured and refocused the introduction to improve clarity for the reader. We also shortened the summary of the presented work in order to provide better and clearer guidance for the subsequent research. - How long can the PJ-based dispensing process keep working with no nozzle clogging occurring? The authors only conducted the test for 45 minutes, but the longer test is also valuable.
Response: The reviewer's thoughtful observation is recognized and appreciated as a valuable contribution to our research. In general, a PipeJet can be operated without restrictions and can print non-stop. Nozzle clogging only occurs when dispensing is paused for several minutes (> 5 minutes), resulting in nozzle drying. In the case of continuous printing of cell suspensions, the platform's mixing module allows unlimited dispensing until the reservoir is emptied (reservoir volume 12 ml). However, it should be noted that although the target substrate is cooled during the printing process, the platform's sample reservoir is maintained at room temperature and normal atmosphere. Therefore, the cells are not maintained in optimal conditions. This means that changes in cell proliferation or spheroid formation are possible the longer the cells are kept outside the incubator. For this reason, we have not evaluated spheroid formation with hanging drop array generation times that exceed 45 minutes. However, further customization of the platform with atmosphere and overall temperature control is possible to allow longer processing times with enhanced cell environment conditions. We have now added this information in section '3.2 Automated large-scale generation of consistent and scalable spheroids'. - The authors should provide the specific efficiency of spheroid generation with different volumes.
Response: We are grateful for the insightful comment provided by the reviewer. We did not observe a drop volume dependent effect on drop printing and spheroid generation efficiency in the range of 1 µl to 4 µl. Out of 15 targeted spheroids, 100% spheroid generation efficiency was observed for each hanging drop volume. For hanging drops ≤ 1.5 µl, it is beneficial to add empty hanging drops (e.g., water) at the drop array boundaries to reduce evaporation effects. We have now added this information in section ‘3.2 Automated large-scale generation of consistent and scalable spheroids’. - The authors expound on spheroid harvesting via centrifugation with an example of a 2 µL hanging drop. How effective is this method for other volumes of spheroids?
Response: We thank the reviewer for this important feedback. We have now added data for spheroid transfer from hanging drops of different sizes. In general, we measured 100% transfer efficiency for hanging drops in the range of 1 µl to 4 µl at centrifugal accelerations of 25 g to 40 g, depending on the drop volume. We also did not see a spheroid size dependent effect on transfer efficiency in that volume range. - In order to visualize the performance of the platform, the author should provide a table comparing the differences between this method and other commonly used methods in terms of production efficiency, dimensional accuracy, cost, etc.
Response: We would like to acknowledge the reviewer's helpful remark, which contributed to enhancing our study. We have now added a table in the Conclusion chapter that compares our novel workflow with common alternative technologies for spheroid generation reported in the literature, such as rotating cell culture flasks, ultra-low attachment plates, hydrogels, or microfluidic chips. The disadvantages of the hanging drop method reported in the literature, namely difficulties in large-scale production, difficulties in long-term culture, high workload and therefore low yield, as well as difficulties in transferring the spheroids, are clearly addressed in our combinatorial workflow of cell suspension mixing during automated hanging drop generation and centrifugal spheroid transfer. This provides significant advantages over other techniques reported in the literature to date. Furthermore, user-driven workflow customization can be easily performed in a standardized and automated manner. This allows for a broad versatility of the presented method for a wide range of applications.
Round 2
Reviewer 3 Report
Comments and Suggestions for Authors
I have no more comments. This paper can be considered for publication in present form.